# Investigation of Material Loading on an Evolved Antecedent Hexagonal CSRR-Loaded Electrically Small Antenna

**DOI:** 10.3390/s23208624

**Published:** 2023-10-21

**Authors:** Jake Peng Sean Ng, Yee Loon Sum, Boon Hee Soong, Paulo J. M. Monteiro

**Affiliations:** 1School of Electrical and Electronic Engineering, Nanyang Technological University, 50 Nanyang Avenue, Singapore 639798, Singapore; ylsum@ntu.edu.sg (Y.L.S.); ebhsoong@ntu.edu.sg (B.H.S.); 2Department of Civil Engineering, University of California, 725 Davis Hall, Berkeley, CA 94720, USA; monteiro@ce.berkeley.edu

**Keywords:** split-ring resonators, electrically small antennas, scattering parameters

## Abstract

Recent advances in embedded antenna and sensor technologies for 5G communications have galvanized a response toward the investigation of their electromagnetic performance for urban contexts and civil engineering applications. This article quantitatively investigates the effects of material loading on an evolved antecedent hexagonal complementary split-ring resonator (CSRR)-loaded antenna design through simulation and experimentation. Optimization of the narrowband antenna system was first performed in a simulation environment to achieve resonance at 3.50 GHz, featuring an impedance bandwidth of 1.57% with maximum return loss and theoretical gain values of 20.0 dB and 1.80 dBi, respectively. As a proof-of-concept, a physical prototype is fabricated on a printed circuit board followed by a simulation-based parametric study involving antenna prototypes embedded into Ordinary Portland Cement pastes with varying weight percentages of iron(III) oxide inclusions. Simulation-derived and experimental results are mutually verified, achieving a systemic downward shift in resonant frequency and corresponding variations in impedance matching induced by changes in loading reactance. Finally, an inversion modeling procedure is employed using perturbation theory to extrapolate the relative permittivity of the dielectric loaded materials. Our proposed analysis contributes to optimizing concrete-embedded 5G antenna sensor designs and establishes a foundational framework for estimating unknown dielectric parameters of cementitious composites.

## 1. Introduction

The deployment of fifth-generation (5G) technology [1,2] since its inception in 2019 has brought about a new era of connectivity, characterized by unprecedented data speeds, enhanced network capabilities, and massive device connectivity in wireless communication. Compared to 4G Long Term Evolution (LTE), 5G is poised to unlock a significantly lower latency rate and higher download speeds as part of 5G performance objectives [3]. The rapid development of 5G has reshaped the landscape of technology and society, primarily driven by advancements in antenna and sensor technologies [4].

Antennas, essentially transducers of electromagnetic (EM) waves, serve as vital conduits that enable the seamless transmission of wireless communication signals. Notably, the realm of antenna technologies is currently undergoing an accelerated phase of research and development efforts, driven by the pursuit of performance optimization, expansion of frequency capabilities, and the adaption to evolving communication demands. This evolution has extended to the integration of antennas with sensor technologies, particularly in applications such as Internet of Things (IoT) [5,6] where antenna sensors are engineered to transmit and receive EM signals while continuously sensing specific parameters of interest in their surroundings such as temperature and humidity.

In wireless sensor networks (WSN) [7], resonator-based electrically small antennas (ESAs) [8] are strategically employed owing to their compactness and effective operation within the designated frequency range. Smart antenna sensors are seamlessly integrated into building structures such as walls, ceilings, and facades, or precast concrete structural elements to ensure minimal disruption to the building’s aesthetics while providing critical information related to structural health, water content, and even aging of building materials over time in the context of modern urban infrastructure.

Although ESAs demonstrate proficiency in achieving reduced physical dimensions for WSN applications, their compactness often leads to compromised performance characteristics in terms of impedance bandwidth and radiation efficiency. As such, achieving antenna miniaturization is challenging for radio frequency (RF) antenna engineers due to fundamental limitations in size and performance governed by the Chu limit [9]. The minimum quality (*Q*) factor is given by:(1)$$Q=\frac{1}{k^{3}a^{3}}+\frac{1}{ka}$$
where $k=\frac{2\pi }{\lambda }$, and *a* is the radius of the hypothetical sphere circumscribing the largest antenna dimension. Since the radius of an ESA is lesser than the radian length (*ka* < 1) as defined by Wheeler [10,11], the minimum *Q*-factor, which is inversely related to the bandwidth-efficiency product, entails tradeoffs in performance. The reduction in electrical size correspondingly leads to lower radiation resistance, poorer radiation efficiency, and a maximum achievable gain of 3 dBi according to the Harrington bound [12].

To optimize the radiation characteristics of ESAs for microwave applications, researchers often employ innovative techniques such as incorporation of metamaterial (MTM) structures [13,14] into the radiating element of the antenna. The split-ring resonators (SRR) introduced by Pendry et al. [15] and its dual, the complementary split-ring resonators (CSRR), introduced by Falcone et al. [16], have successfully demonstrated the synthesis of MTM resonators to achieve antenna miniaturization without sacrificing performance. The SRR, an equivalent of a LC-resonator tank, is excited by a nearby feeding structure through inductive or capacitive coupling and radiates efficiently at the resonant frequency. This class of resonant MTM structures, known as metaresonators, has since become a well-established technique for achieving antenna miniaturization [17] while maintaining desirable performance characteristics from their evolved structures [18]. Their resonance characteristics are characterized by their high *Q*-factor, indicating a narrow frequency bandwidth in which they efficiently operate. Considering that metaresonators can achieve strong resonance at a specific frequency, they are valuable for applications that require precise frequency tuning and selective responses. For the application of lumped circuit model analysis, the maximum dimension of the antenna’s unit cell is limited to a=0.1 λ, where λ is the wavelength corresponding to the operating frequency. With advancements in printed circuit board (PCB) technology, the design of metaresonators into PCBs offers several benefits such as compactness, seamless integration, improved signal integrity, and tailored EM responses.

Prior studies have demonstrated the feasibility of tuning an antenna’s frequency response using material loading techniques [19,20]. This embraced methodology can be considered an analogous framework that provides a basis for studying the EM performance of antennas embedded in concrete [21], especially within the context of WSNs applications. Sum et al. [22] have effectively demonstrated that embedding antenna prototypes into Ordinary Portland Cement (OPC) pastes with varying weight percentages of iron(III) oxide (Fe_2_O_3_) inclusions can perturb the radiating element of a CSRR-loaded antenna operating within the 2.4 GHz Wi-Fi frequency band and induce a change in the loading reactance. The extrinsic modification of the antennas’ operational characteristics was evident through shifts in resonant frequency and corresponding variations in impedance matching, as observed in their reflection coefficient (|S_11_|) parameter plots.

With the advent of 5G, there is a growing motivation to investigate the EM performance of concrete-embedded antenna sensors. Through a multifaceted study, a quantitative investigation is conducted to assess the effects of material loading on an evolved antecedent hexagonal CSRR-loaded antenna design operating in the 3.5 GHz 5G frequency band. The design methodology for the proposed CSRR-loaded ESA is first introduced before delving into the analysis of its operational performance characteristics. Following the conceptualization phase, a physical PCB prototype is fabricated as a proof-of-concept prior to a simulation-based parametric study involving antenna prototypes embedded into OPC pastes with varying weight percentages of Fe_2_O_3_ inclusions (up to 4 wt%). The interoperability of the materially loaded antenna was assessed in two key aspects, specifically resonant frequency and impedance matching, using shift ratios and Δ|S_11_| at resonance as primary metrics. Shift ratios ranging from −5.25% to −16.8%, in tandem with |S_11_| changes, indicate a systemic downward shift in resonant frequency and corresponding variations in impedance matching. Finally, an inversion modeling procedure is employed using perturbation theory to extrapolate the relative permittivity of the dielectric loaded materials from their corresponding shift ratios, achieving a satisfactory fidelity range of 2.04 to 2.27 compared to theoretical predictions. Our proposed analysis paves the way to optimize the EM performance of concrete-embedded antenna sensors operating at mid-band 5G frequencies and establishes a foundational framework for estimating unknown EM parameters of cementitious composites.

This article is organized as follows. Section 2 proposes the design principles for the hexagonal CSRR-loaded ESA, including a comparison between simulated and measured network parameters. Section 3 presents the framework for the material loading mechanism, the concept of perturbation theory, as well as simulation and experimental findings of the designed antenna subjected to material loading. Section 4 details the inversion modeling procedure for extrapolation of the dielectric parameters. The main conclusions drawn in this article are summarized in Section 5.

## 2. Hexagonal CSRR-Loaded ESA Design

### 2.1. CSRR Unit Cell Structure

Figure 1 illustrates the topology of the proposed CSRR unit cell structure for the ESA design and its equivalent-circuit model. Maximum unit cell dimension is selected to be a=0.1λ for the application of lumped circuit model analysis. Geometric configuration of the MTM unit cell is selected to be hexagonal for its tessellating property and the ability to achieve the highest gain relative to its area. This antecedent-based hexagonal-stubbed CSRR structure [23] can be modelled by a shunt *LC*-resonator tank excited by an axial electric field. Exhibiting the behavior of an electric dipole, the CSRR-loaded ESA generates EM waves that propagate along the ring’s surface, consisting of capacitive and inductive elements.

The effective capacitance *C*_0_ is contributed by the aligned split gaps between the pair of stubs [24] at the top of the CSRR structure and the separation distance between the metallic conducting strips while the effective inductance *L*_0_ is contributed by the metal strips of the ring and the two shunted through-hole vias [25] connecting the CSRR to the substrate. To account for the losses and power dissipation, a resistor *R* is added in series as shown in the equivalent-circuit model (see Figure 1b). When the magnitude of the inductive and capacitive reactances are equal, the circuit becomes purely resistive at the resonant frequency $f_{0}$ obtained using (2).
(2)$$f_{0}=\frac{1}{2\pi \sqrt{L_{0}C_{0}}}$$

### 2.2. Design Methodology

Modeling of the proposed CSRR-loaded ESA, disposed on an FR-4 substrate and a copper ground plane, is conceptualized within a simulation environment, where the antenna is designed to operate sufficiently to cover the 3.5 GHz 5G frequency band and resonate desirably at $\lambda_{0}=3.50$ GHz. Compared to other substrates, the selection of FR-4 is driven by its cost-effectiveness, compatibility with standard PCB processes, and ease of precise fabrication. As illustrated in Figure 2, the simulation model is created using Computer Simulation Technology (CST) Microwave Studio Suite^®^.

Symmetrical configuration of both outer and inner SRRs are illustrated, featuring the vertical arrangement of two pairs of stubs positioned at both ends of the aligned split gaps. The length of the stubs is designed as a variable component that can be optimized using the CST optimizer tool to fine-tune the resonant frequency by increasing the trace inductance and mutual coupling between the split rings. Minimum trace separation (*l*_separation_) limit of 0.127 mm is used in accordance with IPC-2221 standards for PCB manufacturing to achieve good coupling between the SRRs. Throughout the design, the thickness of copper traces (*t*_copper_) and ground plane (*t*_ground_) are set to 0.0178 mm while maintaining a trace width (*l*_width_) of 0.127 mm to prevent the change in impedance and signal reflections due to trace discontinuities.

In order to preserve the exterior dimension and shape, an inner SRR is inserted to increase the capacitive coupling. For the outer SRR, two through-hole vias are inserted evenly along both ends of the bottom split where the top and bottom layers are interconnected. Finally, PCB component footprints of 0402 size are created on the bottom layer (see Figure 2b) to serve as feed points or component placement. For simulation purpose, an excitation port of 50 Ω is connected across the edges of the component pads. In order to achieve design optimization, the in-built Classic Powell optimization algorithm is utilized to determine the optimal stub length required for the antenna to achieve resonance at 3.50 GHz. The electrical and physical dimensions of the CSRR-loaded ESA are about $0.1\lambda_{0}\times 0.1\lambda_{0}\times 0.02\lambda_{0}$ and 10.1×10.1×1.6 mm^3^, respectively. Detailed parameter values for the designed antenna are listed in Table 1.

### 2.3. EM Field Distribution Analysis

Using the time-harmonic Maxwell’s curl equations [26], an analysis of the simulated EM field distributions is presented to gain an enhanced understanding of the performance characteristics and behavior exhibited by the optimized antenna during resonance. Figure 3 illustrates visual representations of the generated electric and magnetic fields, along with surface current distributions, within the proposed unit cell at 3.50 GHz.

In response to a time-varying magnetic field, a curl in the electric field arises, as illustrated in Figure 3a by varying color intensity corresponding to the distribution of charged particles along the copper traces. Strong capacitive coupling can be observed between the inner and outer split rings, with the electric field strongly confined within the region delineated by the pair of stubs compared to other regions of the CSRR structure. Based on this observation, it is clear that capacitive coupling mechanism dominates in this configuration as a result of the vertical placement of stubs between the aligned split gaps.

Considering the interrelation between conduction current I and magnetic field B, as defined in (3) with *r* as the permeability of free space, and $\mu_{r}$ as the relative permeability, there is a magnetic field created around the split rings due to the movement of electric charges along the copper traces. From the bilateral symmetrical magnetic field distribution illustrated in Figure 3b, it is evident that the maximum magnetic field intensity occurs in regions of the inner ring above the through-hole vias where the surface current density is the greatest. In contrast to the inner ring, the outer ring exhibits a lower magnetic field intensity due to the decreased mobility of electrical charges along the copper traces.
(3)$$B=\frac{\mu_{0}\mu_{r}I}{2\pi r}$$

Figure 3c illustrates the surface current distributions, which are characterized by the bilateral symmetry. As a result of mutual inductive coupling effects between the split rings, surface current movements are observed to be stronger in regions of the inner ring than the outer ring, which brings about the phenomenon of resonance. Retardation effects in inductive coupling along the copper traces can be deduced from the variations in current density at different portions of the inner ring. For the outer ring, there is a relatively lower current density where the surface current is more uniformly distributed throughout. The relative position and orientation of the elements influence the anisotropy of the CSRR structure, where capacitive and inductive coupling are present.

### 2.4. Simulated Radiation and Network Parameters

Figure 4 shows the simulated 3D radiation pattern plot of the CSRR-loaded ESA, oriented along the *x*- and *y*-axes. At the resonant frequency of 3.50 GHz, the radiation pattern is characterized by a distinctive main lobe and a back lobe extending in the positive and negative *y*-axis direction, respectively. The color intensity corresponds to the gain relative to a hypothetical isotropic antenna radiating uniformly in all directions. The presence of shallow nulls, which are observed traversing the *x*–*z* plane, indicates that the antenna is directional along the boresight.

Figure 5 shows the azimuth and elevation plane radiation pattern plots in polar coordinates. In the azimuth plane (see Figure 5a), the antenna exhibits omnidirectional behavior, characterized by two symmetrical lobes reaching a peak gain of −0.42 dBi. Having an axial ratio of 40 dB (>10 dB), the antenna radiates a linearly polarized wave and achieves a peak gain of 1.80 dBi in the main lobe with a 3-dB angular width of 153.3° in the elevation plane (see Figure 5b). Utilizing the aforementioned electrical parameters of the resonant antenna at 3.50 GHz in Table 1, the maximum achievable gain based on the Harrington bound in (4) is 2.18 dBi, showing satisfactory agreement with the simulated gain of 1.80 dBi. Although the gain of the antenna is relatively low due to its compact size, the gain obtained is still reasonably acceptable.
(4)$$G\left(dBi\right)=10\;log[{\left(ka\right)}^{2}+2(ka)]$$

A logarithmic sweep of the single-port S-parameter measurements was performed from 3.0 to 4.0 GHz using the frequency domain solver. The simulated |S_11_| parameter plot with respect to frequency is presented in Figure 6. The pronounced reflection dip centered at the resonant frequency of 3.50 GHz corresponds to an acceptable |S_11_| value of approximately −20 dB and a voltage standing wave ratio (VSWR) of around 1.22. The operational frequency range of the antenna is established based on the −10 dB impedance bandwidth (BW), followed by the evaluation of the fractional bandwidth (FBW), and *Q*-factor using (5). The simulated network performance parameters are summarized in Table 2. As expected, the designed antenna exhibits an inherent narrowband behavior typical of metaresonators based on the low FBW and high *Q*-factor.
(5)$$Q=\frac{f_{c}}{BW}$$

### 2.5. Measured Network Parameters

In order to establish the fidelity of the simulation model, physical antenna prototypes were fabricated on a PCB, as shown in Figure 7.

Figure 8 shows a 50-Ω RF mini coaxial cable soldered directly onto the component footprint of the antenna’s solder pads. This procedure was carried out prior to the single-port measurement of the S_11_ parameters with the use of a fully calibrated Agilent N5244A PNA-X Network Analyzer (Agilent Technologies, Santa Clara, CA, USA). Using measured |S_11_| data, the associated network performance parameters are similarly evaluated and presented in Table 3.

Figure 9 illustrates the comparison between simulated and measured |S_11_| profiles. The fabricated antenna prototype exhibits resonance behavior at a marginally lower frequency of 3.445 GHz, as opposed to the expected frequency 3.50 GHz optimized through simulation. The trivial deviations in the |S_11_| behavior, along with the subsequent effects on the network performance parameters, can be mainly attributed to the introduction of parasitic capacitance between copper traces and the ground plane as well as static capacitance introduced by the SMA connector, which increases the effective electrical length of the antenna. In comparison to simulated S_11_ results, the relatively low percentage systematic error observed in the experimental results validates the robustness of the simulation model.

### 2.6. State-of-the-Art for Metaresonator Antennas

A comparison between the proposed CSRR-loaded ESA and several state-of-the-art metaresonator antennas is presented in Table 4 to assess the contributions of the antecedent-based design to the field and operational performance characterized by advanced design. The factors considered in this comparison include the shape of the unit cell, physical and electrical dimensions, resonant frequency, and operating frequency bands.

It is noteworthy that the proposed CSRR-loaded ESA and its antecedent design do not demonstrate the characteristic behavior of resonance splitting where multiple resonant frequencies are associated with different modes of EM interaction. Through the adjustment of the size, shape, or orientation of the MTM unit cell, specific resonant frequencies and characteristics can be achieved using SRR(s) from existing research work (see Table 4). The single resonance mode behavior of the proposed CSRR can facilitate the precise control over EM properties for specific applications related to mid-band 5G technology.

## 3. Material Loading on a CSRR-Loaded ESA

In this section, the effects of material loading on a CSRR-loaded ESA are investigated through simulation and experimentation. Firstly, the framework for the material loading mechanism and the research methodology are introduced before the simulation-based designed antenna is subjected to material loading. Next, the concept of perturbation theory is introduced followed by a parametric study conducted on materially loaded antenna prototypes. Subsequently, categorical analysis is performed, and definitive conclusions are drawn.

### 3.1. Material Loading Mechanism

Consider a CSRR-loaded ESA embedded in a non-dispersive lossless medium with relative permittivity $\varepsilon_{r}$, permeability $\mu_{r}$, and electrical conductivity *σ*, wherein the antenna is enclosed within a volume *V* and bounding surface *S*, as schematically illustrated in Figure 10. Assuming that the dielectric loaded material is purely non-magnetic ($\mu_{r}=1$) and non-conductive (σ=0) in a lossless system, the electrical energy input will be equal to the EM energy output based on Foster’s reactance theorem [31]. According to Poynting’s theorem [32] with initial conditions electric field ***E*** (*r*,*t* → −∞) = 0 and magnetic field intensity ***H*** (*r*,*t* → −∞) = 0, the total electrical energy supplied to the antenna system is the sum of EM energy stored in the bounded region, EM energy escaped through the bounding surface, and thermal energy as dielectric losses. The total EM energy *W*_EM_ stored within *V* at time ${t=t}_{0}$ is given in (6).
(6)$$W_{\mathrm{EM}}(t_{0})=\frac{1}{2}\int_{V}^{}\left(\varepsilon_{r}{\left|E\right|}^{2}+\mu_{r}{\left|H\right|}^{2}\right)\;dV$$

.

In a classical lumped circuit model where the input state is voltage **V** and the output state is current **I**, the time-average stored EM energy [33] of the embedded antenna system in the quadratic form expressed in (7) is derived by taking the angular frequency *ω* derivative of the impedance matrix **Z** and multiplying with current **I** matrix and Hermitian conjugate **I**^H^ from the right and left, respectively. The impedance matrix **Z** in (8) can be expressed as the explicit summation of the circuit components, where **R**, **L**, and **C**_i_ represents the resistance, inductance, and capacitance matrix, respectively.
(7)$$W_{EM}=\frac{1}{4}I^{H}LI+\frac{1}{4\omega^{2}}I^{H}C_{i}I$$
(8)$$Z=R+j\omega L+\frac{1}{j\omega }C_{i}$$

Equating the EM energy stored using (6) and (7), it can be established that the capacitance and/or inductance in the embedded antenna system has a dependency on the dielectric properties of the loaded material. For this reason, the loading reactance of the resonant antenna can be adjusted capacitively or inductively to induce a theoretical shift in resonant frequency and subsequently, impacting the overall operational effectiveness of the antenna system.

For a lossy dielectric loaded material, dielectric losses exist within the dielectric medium. These losses can be quantified using (9), where the loss tangent is defined as the ratio of the imaginary part $\varepsilon^{\prime \prime }$ to the real part $\varepsilon^{\prime }$ of the complex permittivity function. From a modeling perspective, the embedded antenna system’s characteristics can be represented by equivalent lossy transmission line. As such, a lower loss tangent of the loaded material can result in an increase in reflected power and consequently, a higher return loss due to improved impedance matching. On this basis, there is an impetus to investigate the implications of the loaded material properties on the network performance parameters of the antenna system in one aspect.
(9)$$\tan\delta =\frac{\varepsilon^{\prime \prime }}{\varepsilon^{\prime }}$$

### 3.2. Methodology

In the pursuit of advancing embedded antenna systems for practical applications in construction and infrastructure, a highly non-intrusive material loading technique is employed in this study. The simulation-based designed antenna is embedded into a dielectric material to replicate real-world conditions where antenna sensors are often concealed within construction materials such as concrete to preserve the aesthetic appeal of structures without compromising the structural integrity. This adopted approach of subjecting the antenna under these emulated conditions can enable the effective measurement of their S_11_ parameters during simulation and experimentation without any detriment to its functionality. Given that concrete-embedded antenna sensors are not conventionally subjected to reuse, they are expected to provide reliable monitoring and communication capabilities throughout the building’s operational lifespan. 

### 3.3. Computational EM Model

In order to gain a preliminary understanding of the EM performance of embedded antenna systems from a theoretical perspective, the simulation-based designed antenna is materially loaded with a dielectric material, as shown in Figure 11. A setting value of the dielectric parameters [34], $\varepsilon_{r}=2.2$ and $\mu_{r}=1.0$ was assigned to replicate concrete-like conditions analogous to the dielectric properties of OPC, a binding agent in concrete manufacturing. Considering the variability in loss tangent for real concrete with differing microstructures and porosity levels [35], the loss tangent parameter is incrementally adjusted in steps of 0.01, spanning from tan⁡δ=0.01 to tan⁡δ=0.10 based on typical loss tangent values by means of a parameter sweep. Through the use of the frequency domain solver, S_11_ parameters are measured across the frequency range of 2.0 to 4.0 GHz prior to the analysis.

Figure 12 illustrates the simulated |S_11_| parameter plots corresponding to different input loss tangent values of the loaded material. A quick observation reveals the variations in the notch depths of the |S_11_| with the preservation of the inherent narrowband behavior for the materially loaded antenna. Upon further analysis of the S_11_ profiles, there is a reciprocal relationship between the notch depth and loss tangent where lower loss tangent values correspond to deeper notch depths, and vice versa. A higher (more negative |S_11_|) return loss value is desirable and typically indicates a well-matched system with efficient energy transfer and minimum signal reflections. Conversely, a lower (close to 0 dB) return loss value indicates poor impedance matching and high signal reflections, leading to reduced efficiency and undesirable performance.

In addition to achieving a maximum return loss of 38.6 dB, there is an observable shift in resonant frequency after the designed antenna was subjected to material loading. The interaction between the EM properties of the loaded material and the antenna’s electric and magnetic fields changes the effective electrical properties of the materially loaded antenna to induce a downward shift from 3.50 GHz to a transposed resonant frequency *f*_1_ of about 3.036 GHz. Based on fundamentals of EM theory, there is a change in loading reactance of the embedded antenna system wherein the loaded material introduces parasitic capacitance by creating an electric field and storing charges on their surfaces during the antenna operation. Due to perturbations in the EM field distribution and surface currents within the antenna structure, this in turn affects the frequency response and operating regime of the resonant antenna. Equivalent-circuit model of the embedded antenna system is consequently derived in Figure 13 with the parallel addition of the capacitance *C*_c_ to the established CSRR-loaded equivalent circuit.

### 3.4. Perturbation Theory

The effects of material loading on the system behavior can be effectively demonstrated through the application of perturbation theory [36]. Assuming a Hermitian system for which EM energy is conserved and no changes in permeability (Δµ=0), the shift ratio (S.R.) of the resonant frequency is predicated by (10) which quantifies local changes in permittivity (Δ*ε*) within the specified *V* in response to the electric and magnetic field intensities denoted as $E_{0}$ and $H_{0}$. Alternatively, the S.R. can also be determined in terms of stored energies by integrating the relative change in electric energy density ${\bar{w}}_{e}$ within the specified volume *V* using (11). In practical terms, a high S.R. indicates a significant change in permittivity within the system, while a low S.R. suggests a system with an inherently stable EM behavior. As postulated by perturbation theory, the increase in permittivity is positively correlated with a capacitive effect, which has been empirically confirmed by the downward shift in resonant frequency corresponding to a S.R. of about −13.3% from simulation results. For this reason, the S.R. provides a quantitative measure of how changes in the dielectric parameters of the loaded material influence the distribution of the EM energy within a given region and the antenna performance therewith.
(10)$$S.R.=\frac{f_{1}-f_{0}}{f_{0}}\approx -\frac{{∭}_{V}^{}∆\varepsilon \left|{E_{0}}^{2}\right|\;dV}{{∭}_{V}^{}\left(\mu \left|{H_{0}}^{2}\right|+\varepsilon \left|{E_{0}}^{2}\right|\right)\;dV}$$
(11)$$S.R.\approx -\frac{1}{W_{EM}}{∭}_{V}^{}\frac{∆\varepsilon }{\varepsilon }.{\bar{w}}_{e}\;dV$$

As established by simulation utilizing the Finite-Difference Time-Domain (FDTD) method, which solves Maxwell’s equations numerically on a discretized grid, it is apparent that the shift in resonant frequency has a dependency on the change in permittivity of the loaded material. Although the microwave-material interaction is rigorously accounted for, the FDTD method is restricted to non-dispersive or lossy materials with known EM parameters, which are frequency-independent. For cementitious composites, they are often heterogeneous in their interior and their effective relative permittivities are dependent on the intrinsic dielectric parameters of the host matrix and inclusions, volumetric fractions, and morphology related to shapes and sizes of the inclusions. As such, a semi-empirical approach is required to achieve a further understanding of the effects of material loading, which will be elaborated in the next sub-section.

### 3.5. Experimental Program

In the scope of this parametric study, antenna prototypes are subjected to material loading following the prescribed limit of 4 wt% Fe_2_O_3_ in the cement paste, as outlined in Table 5. This selection is strategically chosen to closely represent the typical amount of reddish-brown admixture pigments used for improving the aesthetics of hardened concrete in accordance with ASTM C979 standards [37]. The water-to-cement (*w*/*c*) ratio was kept at 0.50 throughout to achieve good compressive strength and maintain the degree of saturation based on industry practice and standards for concrete casting. For a well-balanced experimental design [38], a good spacing of the factor levels (weight percentage of Fe_2_O_3_) with a step size of 1 wt% is selected to optimize the number of samples per treatment group as well as to reduce the sensitivity and potential measurement errors on experimental results.

Using 2 replicates per treatment group, 10 antenna prototypes (see Figure 14), each fed by a 50-Ω RF mini coaxial cable, were selected for the experimental campaign based on their comparable S_11_ profiles. The casting process began by mixing the dry constituent(s) for 90 s before adding water, followed by a continuous mixing for 5 min until a homogeneous state was achieved. The materially loaded antenna samples were cast in batches using precast molds of commensurate dimensions such that the samples were completely immersed within a uniform volume of cement paste. A specific depth of 15 mm cement paste was selected to ensure coverage of the antenna’s radiating near-field region (≤1.71 mm) during resonance.

Uniform curing of the samples was carried out under laboratory conditions by covering them with cling film as a non-intrusive method to prevent excessive moisture loss and reduce susceptibility to cracks. To achieve the desired material properties of the hardened cement paste upon curing, ambient conditions were maintained across all samples during the hydration process with relative humidity kept within the range of approximately 40% to 45%. The embedment process was completed with the mold removal followed by a 30-day curing period to allow the hardened cement paste to reach a high degree of curing (approximately 99% of hydration process completed). The schematic and actual materially loaded antenna prototypes (after demolding) are shown in Figure 15.

### 3.6. S_11_ Measurements and Results

In line with previous measurements, the source of excitation from Port 1 of a fully calibrated Agilent N5244A PNA-X Network Analyzer was set to −3 dBm as the output power and 0 dBm as the reference power before a logarithmic sweep of the S_11_ parameter measurements ranging from 2.70 to 3.70 GHz (frequency of interest) was performed for each of the material-under-test (MUT). The schematic and actual S_11_ measurement platform diagrams for the single-port test are presented in Figure 16. Quality of the contemporaneously measured S_11_ data was ensured through a series of repeatable measurements where the measurement system exhibits a high degree of accuracy and reliability of less than 1 dB.

Figure 17 presents the categorical |S_11_| profiles of the antenna prototypes both before and after being subjected to material loading. Notably, there is a systemic downward shift in resonant frequency from $f_{0}$ to $f_{1}$ for all of the materially loaded antennas, showing a generally good agreement with the simulation-derived results. Further observation of the |S_11_| profiles reveals an overall relatively low degree of heterogeneity between MUT samples from the same treatment group. However, one may observe some heterogeneity in the resonant frequency shifts for samples #5A and #5B (see Figure 17e) across the measured frequency range. Such behavior are not due to accuracy of the measurement system but indeed attributed to inherent differences in the microstructure and porosity levels of the hardened cement paste where complex chemical and physical processes occur during its formation and curing. Another potential confounding variable, stemming from alterations in the impedance characteristics, might have also contributed to the observed heterogeneity in the frequency response of the materially loaded antennas. 

In order to assess the interoperability of the materially loaded antennas, the S.R. and Δ|S_11_| at resonance were evaluated for all MUT and presented in Table 6. Given the preservation of the overall shape in the S_11_ profiles, the BW and *Q*-factor will not be evaluated in the interest of brevity.

With reference to Table 6, the negative S.R. across all MUT samples affirm the consistent downward shift in the resonant frequency after subjecting the antenna prototypes to material loading. Moreover, significant variations in the S.R. were observed, illustrating the sensitivity of the resonant frequency shift to varying weight percentages of Fe_2_O_3_ inclusions within the embedding material. In the case of samples #1A and #1B, which are deficient in Fe_2_O_3_ (0 wt%), their S.R. of −5.51% and −5.25%, respectively, are relatively lower in comparison to the S.R. range of −16.8% to −13.9% in the other MUT samples with Fe_2_O_3_ inclusions. Further analysis of the S_11_ profiles and their corresponding S.R. reveals a clear inverse correlation between the S.R. and the weight percentages of Fe_2_O_3_ (ranging from 1 wt% to 4 wt%) where a general diminishing trend in the S.R. was observed across the range. In the S_11_ profiles of samples #2A and #2B, a considerably pronounced downward shift in their resonant frequencies was apparent, with both featuring corresponding S.R. of about −16.8%. Overall, there is a capacitive shift in the loading reactance of the materially loaded antenna. 

Last but not least, a fitting curve was generated for resonant frequencies corresponding to different Fe_2_O_3_ weight percentages in the embedding material, as a means to model the relationship between the two variables as well as to assess the variability associated with each measurement. Two additional S_11_ measurements were performed for each MUT, and the basis for fitting the curves was determined by selecting the resonant frequency value at the 50% quantile from the two MUT samples within the same treatment group, as illustrated in Figure 18. Error bars were incorporated in the fitting curve to indicate the range between the minimum and maximum values. Consistent measurements were achieved for MUT samples in the 2 wt% Fe_2_O_3_ treatment group, whereas MUT samples in the 4 wt% Fe_2_O_3_ treatment group exhibited variations in resonant frequency of up to 0.08 GHz, as previously elucidated. In summary, the measurement campaign, backed by rigorous analysis and robust methodologies, provides a comprehensive investigation of the effects of material loading on a CSRR-loaded ESA.

## 4. Extrapolation of Dielectric Parameters

In this section, an inversion modeling procedure is proposed based on perturbation theory to numerically extrapolate the relative permittivity of the various dielectric embedding materials. Dielectric loss tangent values are also qualitatively extrapolated based on simulation-derived results. The extrapolated parameters are systematically analyzed to provide a comprehensive understanding of the EM characteristics for the embedding materials.

### 4.1. Perturbation Theory-Inspired Modeling

Our modeling approach is inspired and guided by the fundamental principles of perturbation theory, which provide a framework for understanding and quantifying the effects of small changes or perturbations within complex systems. The analytical expression in (11) serves as the mathematical embodiment to establish a direct relationship between the S.R. and changes in permittivity (Δ$\varepsilon_{r}$) relative to its initial value ($\varepsilon_{r,initial}$) within a specified volume *V.*

In the context of this study, simulation parameters of the material-perturbed antenna, including relative permittivity and S.R., were regarded as baseline values to provide a reference point for evaluating $\varepsilon_{r,initial}$ of the unperturbed antenna prototypes under the presumption that their permittivity values within the system were both homogeneous and characterized by negligible dielectric losses ($\varepsilon_{r,initial}^{\prime \prime }=0$). 

Having established these initial conditions, the S.R. for each of the material-perturbed antennas is evaluated based on the shift in resonant frequency relative to its initial unperturbed state. The determined S.R. values, as presented in Table 6, serves as the key parameter for the extrapolation procedures in modeling the linear frequency shift such that there is a direct proportionality between the S.R. and Δ$\varepsilon_{r}$ relative to $\varepsilon_{r,initial}$, as determined previously. Subsequently, $\Delta \varepsilon_{r}$ for each of the MUT sample is extrapolated from the corresponding S.R. values followed by the absolute relative permittivity, as presented in Table 7. A summary of the extrapolation procedures can be found in the flowchart presented in Figure 19.

### 4.2. Discussion

Definitive conclusions can be drawn from the extrapolated parameters for the various dielectric embedding materials. First, there is a quantitative relationship between the relative permittivity and the chemical composition of the dielectric embedding material, where OPC is the primary constituent. The prevalent positive increase in permittivity is primarily driven by the presence of lime, silica, and other metallic oxides in OPC, which facilitates the storing of electrical energy for a given voltage during the antenna operation. Second, the addition of Fe_2_O_3_ inclusions to the embedding material has considerably enhanced the permittivity in comparison to MUT samples (#1A, #1B) which are deficient in Fe_2_O_3_ inclusions (0 wt%). By means of the creation of an electric field that allows the accumulation of additional charges on their surfaces during the antenna operation, this results in a capacitive change in the loading reactance and consequently, a greater downward shift in the resonant frequency.

Upon further investigation, one may observe a peripheral decrease in the relative permittivity (see Table 7) as the weight percentage of Fe_2_O_3_ increases from 1 wt% to 4 wt% in the dielectric embedding material. This phenomenon can be attributed to the dielectric losses introduced by the Fe_2_O_3_ inclusions in a real cementitious composite, suggesting that the permittivity function is not real-valued but complex in nature $\left({\varepsilon_{r}=\varepsilon }^{\prime }-j\varepsilon^{\prime \prime }\right).$ In regard to the loaded antenna characteristics, the magnitude of the complex permittivity |$\varepsilon_{r}$| has an effect on the behavior of EM fields, which consequently lead to varying degrees of shifts in resonant frequency and variations in impedance matching as previously discussed. The real permittivity $\varepsilon^{\prime }$ signifies the amount of stored electric energy, which primarily affects the material’s polarization and capacitive shift in loading reactance. In contrast, the imaginary component $\varepsilon^{\prime \prime }$ signifies the dielectric losses associated with energy dissipation, leading to a decreased amplitude of the reflected wave and affecting the impedance matching between the antenna and its surrounding environment. As the weight percentage of Fe_2_O_3_ increases from 1 wt% to 4 wt%, |$\varepsilon_{r}$| decreases due to the dominant effect of $\varepsilon^{\prime \prime }$ where reduction in dielectric losses outweighs the increase in real permittivity due to Fe_2_O_3_ inclusions. The densely packed extrapolated values ranging from 2.04 to 2.27 for the cementitious composites are verified to fall within the expected bounds [35] of the relative permittivity envelope, which approximately spans from 2.0 to 3.0. This validation highlights a satisfactory degree of fidelity observed when comparing experimental data with theoretical predictions.

Henceforth, the variations in the loss tangent of the dielectric embedding materials are investigated. Since the loss tangent is the controlling factor for the depth of notches in the |S_11_| parameter plots, as hypothesized, evaluations will be based on |S_11,after_| values measured at resonance (refer to Table 6) and simulation-derived results (refer to Figure 12). Accordingly, expected bounds of the loss tangent envelope are estimated to be within the range of tan⁡δ  = 0.01 to 0.02, indicating tightly clustered loss tangent values between MUT samples. For this reason, the loss tangent is categorized into nominal (low/high) ranges to facilitate a more meaningful interpretation of the data and enable the discrimination of the dielectric properties based on their chemical composition. For a low relative loss tangent, a threshold setting value of at least 3 dB change (Δ|S_11_| > 3 dB) is selected, in addition to achieving a higher return loss after subjecting the antenna to material loading. An extrapolated relative loss tangent of the MUT samples is presented in Table 6, where the MUT samples with higher weight percentages of Fe_2_O_3_ (3 wt% and 4 wt%) or a deficiency in Fe_2_O_3_ (0 wt%) exhibit low relative loss tangents. This enables the embedding material to effectively act as an attenuator and lead to an increase in reflected power as a result of improved matching between the input impedance of the ESA and the hexagonal-stubbed CSRR output impedance at resonance.

Conversely, the dielectric embedding material is considered to have a high relative loss tangent, as exhibited by MUT samples with lower weight percentages of Fe_2_O_3_ (1 wt% and 2 wt%). As such, a mild degradation or uniformity in impedance matching can be observed for samples #2A, #2B, #3A, and #3B, as indicated by their S_11_ profiles (see Figure 16b,c).

## 5. Conclusions

This article has successfully investigated the effects of material loading on an evolved antecedent-based design of a hexagonal-stubbed CSRR-loaded ESA through simulation and experimentation. The design principles for the proposed ESA are presented and complemented by an extensive analysis of the EM field distribution to ensure that the antenna design aligns with the performance expectations, wherein the optimized stub length has indeed resulted in resonance at the desired frequency of 3.50 GHz. As demonstrated by the |S_11_| parameter plots, variations in the chemical composition and dielectric parameters of the embedding materials can have a considerable influence on the frequency response, specifically in terms of resonant frequency shift and impedance matching, which are the two key aspects focused on in this research work.

In addition, perturbation theory is adopted in an inversion modeling procedure to numerically extrapolate the relative permittivity of the dielectric embedding materials based on corresponding shift ratios. Dielectric loss tangents were also qualitatively extrapolated to establish a causal relationship between the notch depth in the |S_11_| parameter plots and impedance matching based on simulation-derived results.

Future research can consider incorporating other aggregate types such as fine-sand or fly-ash cenospheres in cementitious composites and compare the different effects of material loading on a resonant antenna. Depending on the intended application, material loading offers the potential as an antenna miniaturization technique to flexibly tune the antenna’s resonant frequency to align with the desired resonant frequency. This approach eliminates the need for employing techniques such as impedance matching networks to overcome inherent performance limitations. In the context of concrete-embedded antenna sensors, alterations in the resonant frequency shift can introduce measurement errors during the continuous sensing of specific surrounding parameters such as temperature and humidity measurements. As such, calibration and correction algorithms are crucial to account for these shifts and ensure that the sensor provides reliable environmental measurements. Moreover, the utilization of material loading techniques with perturbation theory can provide a foundational framework for estimating the unknown EM parameters of cementitious composites.

Overall, this research endeavor highlights the implications of building materials on the EM performance of concrete-embedded antenna sensors. Our proposed analysis can help optimize the design of concrete-embedded antenna sensors for mid-band 5G frequencies, considering the inherent shifts in resonant frequency and adjustments in impedance matching upon embedment.

## Figures and Tables

**Figure 1 sensors-23-08624-f001:**
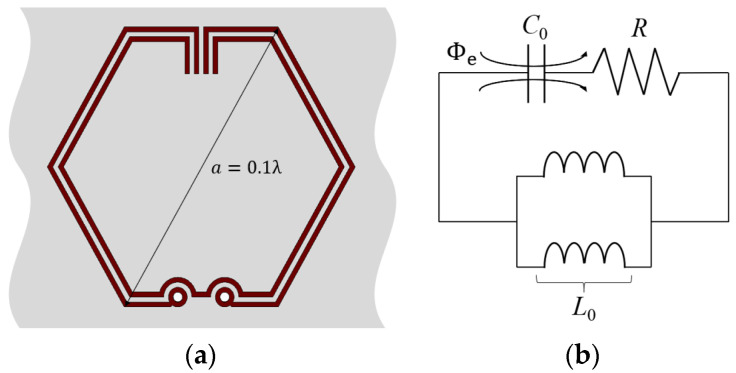
Proposed CSRR unit cell structure: (**a**) Topology; (**b**) Equivalent-circuit model.

**Figure 2 sensors-23-08624-f002:**
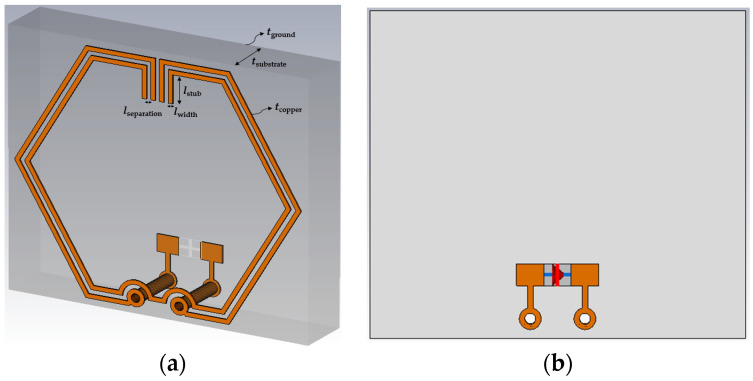
Simulation model: (**a**) Perspective view; (**b**) Back view.

**Figure 3 sensors-23-08624-f003:**
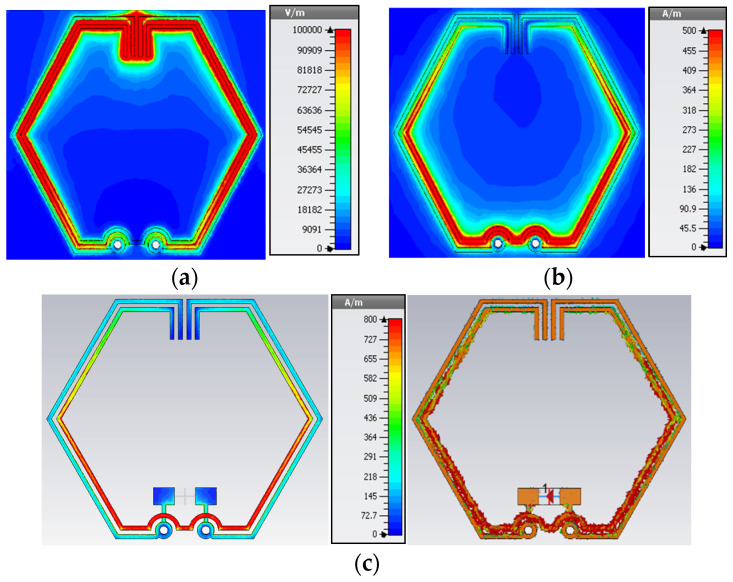
EM field distributions: (**a**) Electric field; (**b**) Magnetic field intensity; (**c**) Surface current.

**Figure 4 sensors-23-08624-f004:**
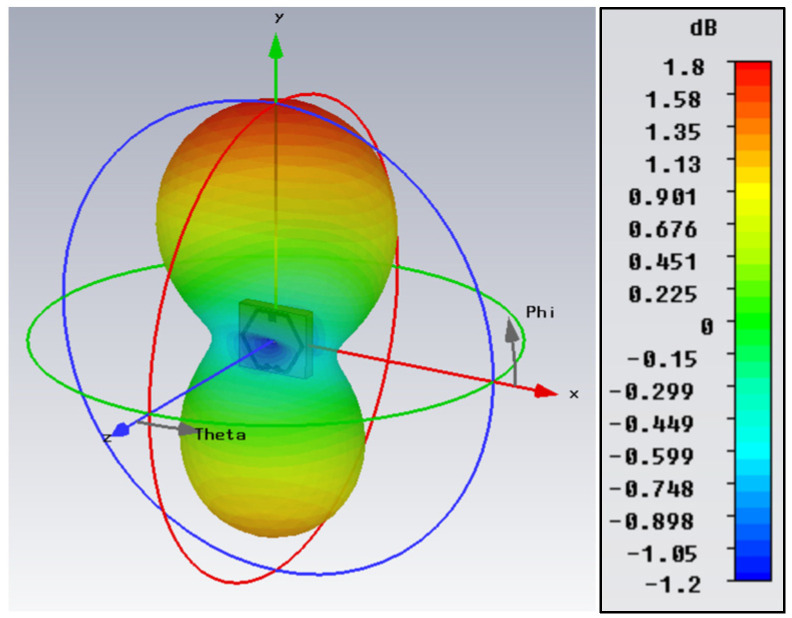
Simulated 3D radiation pattern plot at 3.50 GHz.

**Figure 5 sensors-23-08624-f005:**
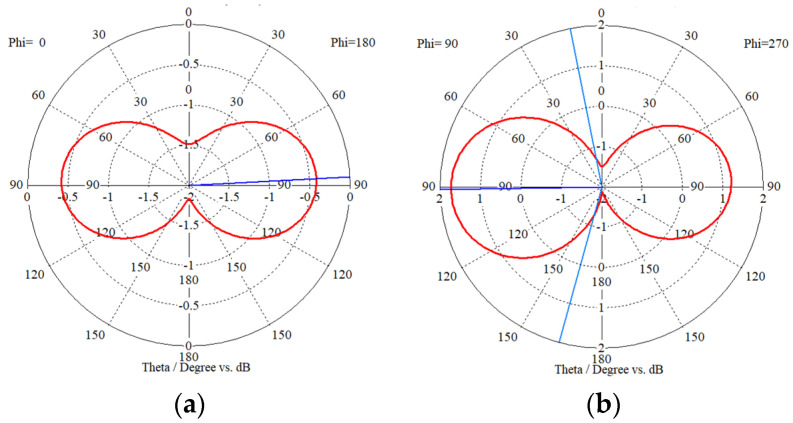
Simulated 2D radiation pattern plots at 3.50 GHz: (**a**) Azimuth plane (*x–z* plane); (**b**) Elevation plane (*y–z* plane).

**Figure 6 sensors-23-08624-f006:**
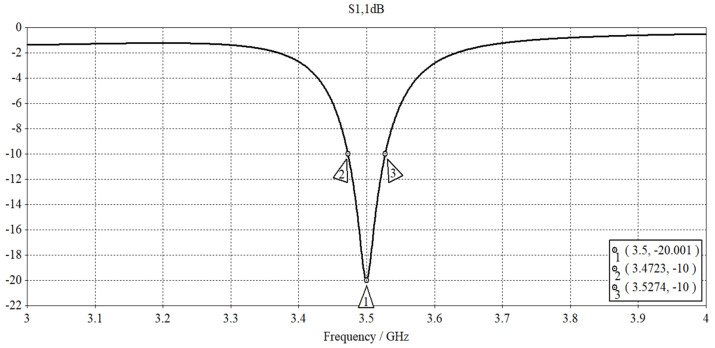
Simulated |S_11_| vs. frequency.

**Figure 7 sensors-23-08624-f007:**
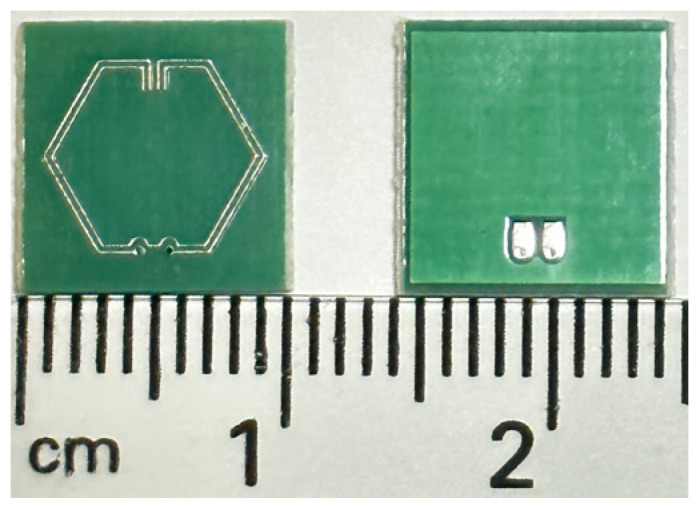
Front and back of antenna prototypes fabricated on a PCB.

**Figure 8 sensors-23-08624-f008:**
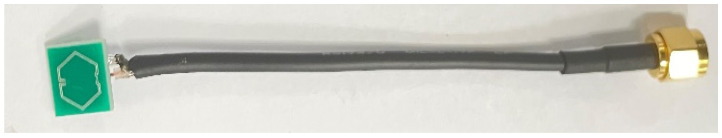
Antenna prototype fed by a 50-Ω RF mini coaxial cable.

**Figure 9 sensors-23-08624-f009:**
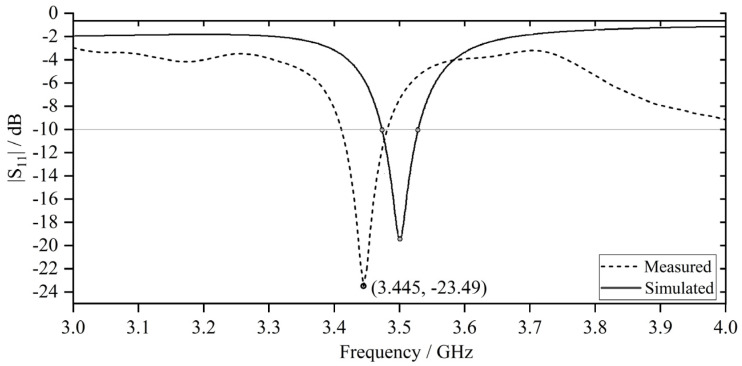
Comparison between simulated and measured |S_11_| vs. frequency.

**Figure 10 sensors-23-08624-f010:**
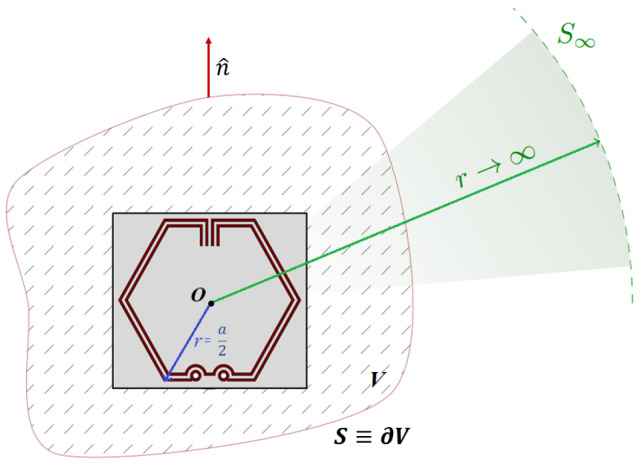
Schematic of an embedded CSRR-loaded ESA circumscribed by a sphere of radius *a* and far-field sphere bounded by $S_{\infty }$.

**Figure 11 sensors-23-08624-f011:**
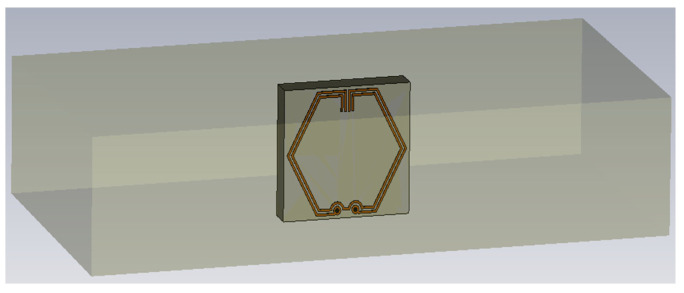
Simulation model of CSRR-loaded ESA in embedding medium.

**Figure 12 sensors-23-08624-f012:**
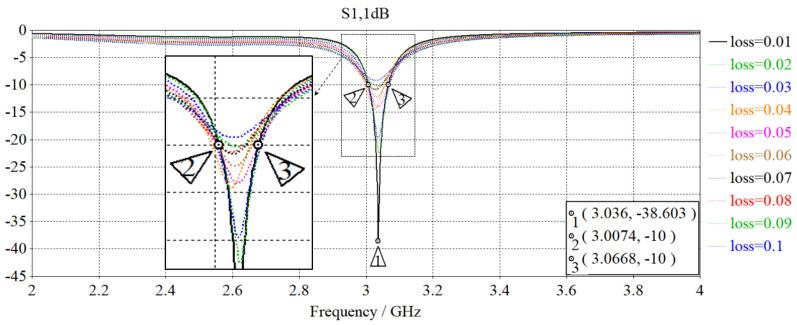
Comparison of simulated |S_11_| vs. frequency for different loss tangent values.

**Figure 13 sensors-23-08624-f013:**
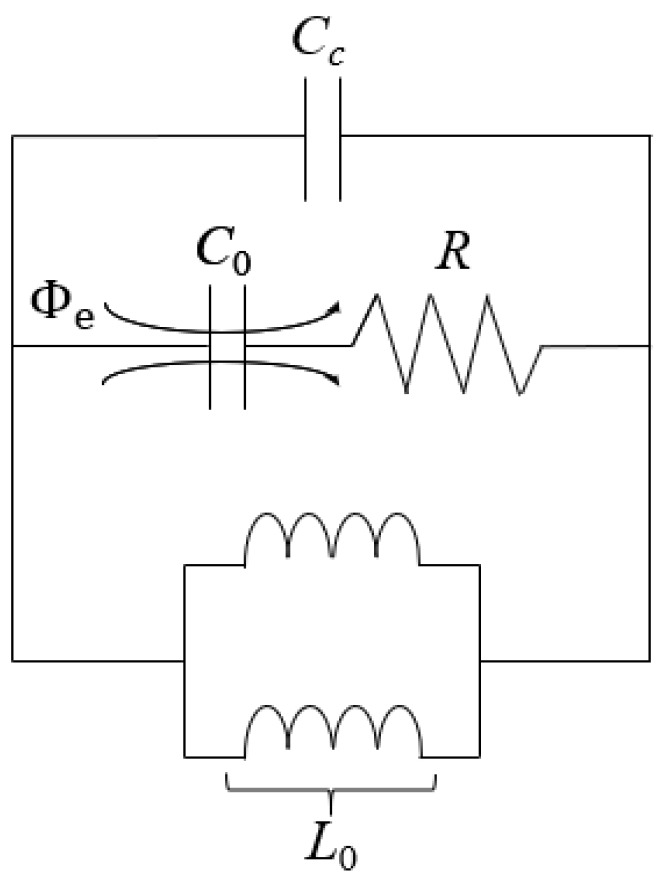
Equivalent-circuit model for embedded antenna system.

**Figure 14 sensors-23-08624-f014:**
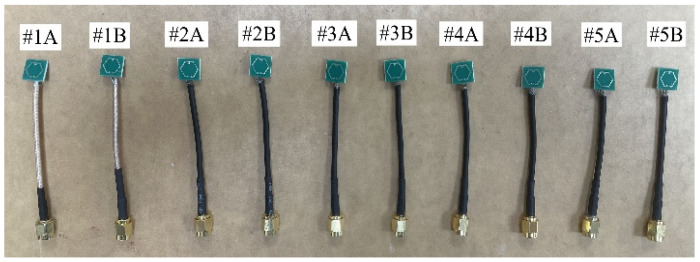
Fabricated antenna prototypes fed by an RF mini coaxial cable.

**Figure 15 sensors-23-08624-f015:**
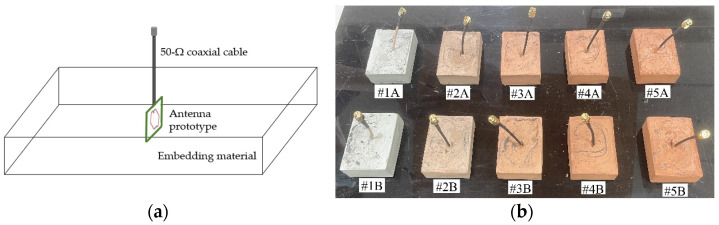
Materially loaded antenna prototype: (**a**) Schematic; (**b**) Actual (after demolding).

**Figure 16 sensors-23-08624-f016:**
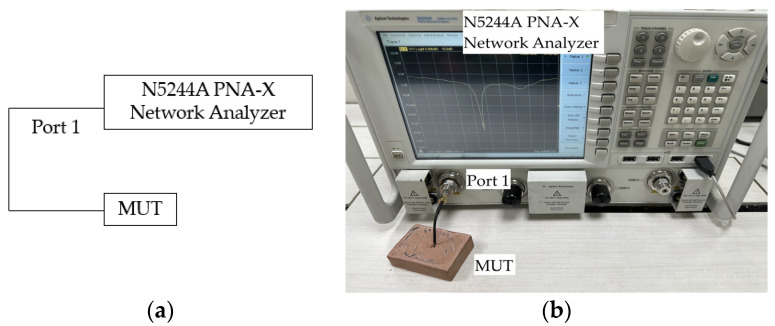
S_11_ measurement platform diagram: (**a**) Schematic; (**b**) Actual.

**Figure 17 sensors-23-08624-f017:**
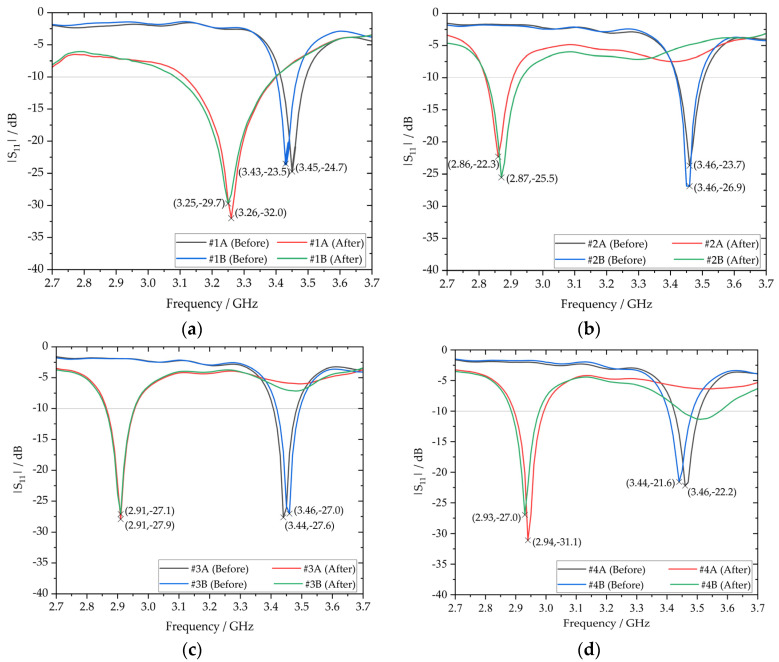
Categorical |S_11_| profiles before and after embedment of antenna prototypes: (**a**) 0 wt%; (**b**) 1 wt%; (**c**) 2 wt%; (**d**) 3 wt%; (**e**) 4 wt% Fe_2_O_3_ inclusions.

**Figure 18 sensors-23-08624-f018:**
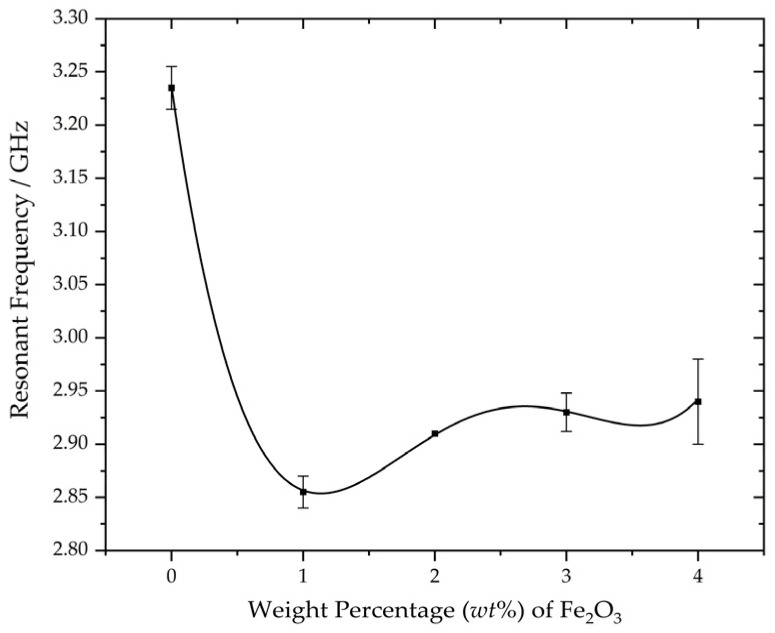
Fitted curve for variation in resonant frequencies for different weight percentages of Fe_2_O_3_.

**Figure 19 sensors-23-08624-f019:**
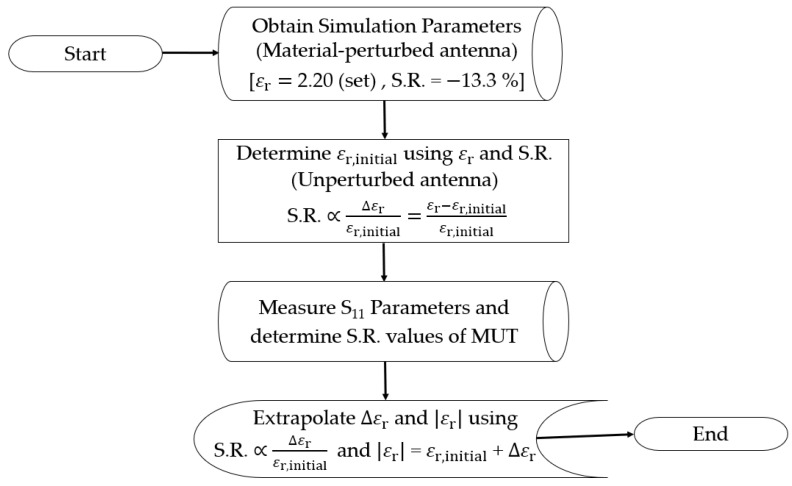
Summary of the extrapolation procedures.

**Table 1 sensors-23-08624-t001:** List of parameter values for the designed antenna.

Electrical and Design Parameters	Symbol	Value
$\mathrm{Maximum}\;\mathrm{cell}\;\mathrm{dimension}\;(\lambda_{0}=3.50$ GHz)	$a=0.1\lambda_{0}$	8.57 mm
Wavelength number	*k*	0.0733 rad/mm
Electrical size	*ka*	0.628
Substrate thickness	*t* _substrate_	1.60 mm (63 mils)
Ground plane thickness	*t* _ground_	0.0178 mm (0.5 oz)
Copper thickness	*t* _copper_	0.0178 mm (0.5 oz)
Width of copper trace	*l* _width_	0.127 mm (5 mils)
Trace separation/clearance	*l* _separation_	0.127 mm (5 mils)
Through-hole via inner radius	*l* _radius,in_	0.127 mm (5 mils)
Annular ring outer radius	*l* _radius,out_	0.254 mm
Optimized stub length	*l* _stub_	0.83 mm
Trace characteristic impedance	-	50 Ω

**Table 2 sensors-23-08624-t002:** Simulated network performance parameters.

$f_{0}$/GHz	|S_11_|/dB	VSWR	BW/GHz	FBW/%	*Q*
3.50	−20.0	1.22	0.055	1.57	63.5

**Table 3 sensors-23-08624-t003:** Measured network performance parameters.

$f_{0}$/GHz	|S_11_|/dB	VSWR	BW/GHz	FBW/%	*Q*
3.445	−23.49	1.15	0.07	2.03	49.2

**Table 4 sensors-23-08624-t004:** Comparison between proposed design and state-of-the-art metaresonator antennas.

Shape of Unit Cell	Physical Dimension/mm^2^ Electrical Dimension/λ^2^	$f_{0}$/GHz	Frequency bands	Reference
Double L-shaped SRR	10 × 10 0.25λ × 0.25λ	7.69, 8.46, 13.12, 14.03	C-, X-, and Ku-bands	[27]
Hexagonal gap coupled SRR	10 × 10 0.11λ × 0.11λ	3.55, 11.80	S- and X-bands	[28]
Coupled ring SRR	8 × 8 0.064λ × 0.064λ	2.24, 4.77, 5.94, 8.94, 10.84	S-, C-, and X-bands	[29]
Octagonal spider net-shaped triple SRR	10 × 10 0.09λ × 0.09λ	2.86, 4.28, 6.94, 9.22, 11.10, 12.89	S-, C-, X-, and Ku-bands	[30]
Hexagonal CSRR (Antecedent)	13 × 13 0.1λ × 0.1λ	2.442	Wi-Fi band	[23]
Hexagonal CSRR (Evolved)	10 × 10 0.1λ × 0.1λ	3.50	Mid-band 5G	This work

**Table 5 sensors-23-08624-t005:** Compositional parameters and mix design for the casting of cement paste.

Parameter/Constituents	Level(s)	Mass of Constituents (kg/m^3^)	Sample #
Cement type (control)	OPC CEM I	1222	-
Water (control)	*w*/*c* = 0.50	611	-
Weight percentage of Fe_2_O_3_	0 wt%	0.0	#1A, #1B
	1 wt%	12.2	#2A, #2B
	2 wt%	24.4	#3A, #3B
	3 wt%	36.6	#4A, #4B
	4 wt%	48.8	#5A, #5B

**Table 6 sensors-23-08624-t006:** Evaluation of S.R. and Δ|S_11_| at resonance.

MUT	$f_{0}$/GHz	$f_{1}$/GHz	Δ*f*/GHz	S.R.	|S_11,_ _before_|/dB	|S_11,_ _after_|/dB	Δ|S_11_|/dB
#1A	3.45	3.26	−0.19	−5.51%	−24.7	−32.0	−7.3
#1B	3.43	3.25	−0.18	−5.25%	−23.5	−29.7	−6.2
#2A	3.46	2.88	−0.58	−16.8%	−23.7	−23.2	+0.5
#2B	3.45	2.87	−0.58	−16.8%	−26.9	−25.5	+1.4
#3A	3.44	2.91	−0.53	−15.4%	−27.6	−27.9	−0.3
#3B	3.46	2.91	−0.55	−15.9%	−27.0	−27.1	−0.1
#4A	3.46	2.94	−0.52	−15.0%	−22.2	−31.1	−8.9
#4B	3.44	2.93	−0.51	−14.8%	−21.6	−27.0	−5.4
#5A	3.44	2.92	−0.52	−15.1%	−28.1	−33.8	−5.7
#5B	3.46	2.98	−0.48	−13.9%	−28.6	−36.7	−8.1

**Table 7 sensors-23-08624-t007:** Extrapolated parameters for various dielectric embedding materials.

MUT	Dielectric Embedding Material	Δ$\varepsilon_{r}$	$|\varepsilon_{r}$|	tan⁡δ (Relative)
-	Simulation-defined	+0.26	2.20	-
#1A	OPC/0 wt% Fe_2_O_3_	+0.11	2.05	Low
#1B	OPC/0 wt% Fe_2_O_3_	+0.10	2.04	Low
#2A	OPC/1 wt% Fe_2_O_3_	+0.33	2.27	High
#2B	OPC/1 wt% Fe_2_O_3_	+0.33	2.27	High
#3A	OPC/2 wt% Fe_2_O_3_	+0.30	2.24	High
#3B	OPC/2 wt% Fe_2_O_3_	+0.31	2.25	High
#4A	OPC/3 wt% Fe_2_O_3_	+0.29	2.23	Low
#4B	OPC/3 wt% Fe_2_O_3_	+0.29	2.23	Low
#5A	OPC/4 wt% Fe_2_O_3_	+0.29	2.23	Low
#5B	OPC/4 wt% Fe_2_O_3_	+0.27	2.21	Low

## Data Availability

Not applicable.

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
