# Peer review of "Investigation of Material Loading on an Evolved Antecedent Hexagonal CSRR-Loaded Electrically Small Antenna"

_sensors, 2023, doi:10.3390/s23208624_

Round 1
Reviewer 1 Report
This article quantitatively investigates the effects of material loading on an evolved antecedent-based design of a hexagonal-stubbed Complementary Split-Ring Resonator (CSRR)-loaded antenna through simulation and experimentation. Simulation-derived and experimental results are mutually verified, achieving a systemic downward shift in resonant frequency and corresponding variations in impedance matching induced by changes in the loading reactance. I have a few concerns while reading the manuscript.
1. There is too much content written in Introduction, and when introducing the research background, it is often not necessary to explain it in such a large amount of space. The author needs to summarize and refine it.
2. In the “Extrapolation of Dielectric Parameters”, the author uses the complex dielectric constant (?r=?’-j?’’) to explain the relative dielectric constant change caused by the weight percentage increasing from 1 wt% to 4 wt% in the embedded material. Please elaborate in detail. Research has shown that the real and imaginary parts of the dielectric constant are related to the resonant frequency and amplitude of the S parameter. Please discuss the relevant connections based on the results of this experiment.
3. The fitting curves of resonance and MUT should be added, and error bar is denoted calculated with 3 times measurements at least.
4. Firstly, the author needs to provide a complete measurement platform diagram, however, Figure 15 is a schematic diagram of the measurement. The author did not describe the depth at which the device penetrates into the cement mold. Does this depth meet the radiation range of the device? The author needs to explain this question. Secondly, the author also needs to explain the impact of the device measurement results proposed for measuring environmental temperature and humidity.
5. The author uses FR-4 as substrate of structure of CSRR antenna, why is this substrate used? In other words, what are the advantages of the substrate used by the author compared to other substrates.
6. In sections 2.3 and 2.4, various simulations were conducted on the designed devices, including electric field, magnetic field, surface current density, and radiation field distribution. What is the significance of these simulation results? Please describe what advantages these results represent for the proposed device or what impact they will have on device performance. Formulas (2) - (3) do not seem to reflect the relationship between the properties of the tested object and the measurement results.
7. In Figure 15 (b), the volume of each test sample seems to be inconsistent, and it can be clearly observed that samples # 1A and # 5B are not consistent. Will this difference lead to a deviation in the results? In addition, what is the purpose of using cling film to prevent sample dehydration during sample preparation, and will the moisture content of the sample affect the results?
8. Why choose to nest the device inside the sample as a measurement method, and whether the device has the conditions for reuse. It should be explained in conjunction with practical application scenarios.
9. Is the resonance frequency response the same after a long time. That is to say, whether the measurement results after three days or one week are the same as the initial measurement results, and if they are different, please explain the relevant reasons.
10. The picture in Figure 1 is not clear enough, and it is recommended to mark the dimensions in Figure 1a. The pictures in the text are not clear enough, and some pictures are blurred, such as Figure 3, Figure 4, Figure 5, etc. The cut-off frequencies for the simulation and measurement are different in Figure 9. The S11 parameters of different materials do not change significantly, and it is recommended that Figure 12 be enlarged in the range of 2.8-3.2Ghz. In Figure 16, the S11 parameters at the resonant frequency are hard to get the details because different results are too close to differentiate. So it should be enlarge to figure out the frequency shift and the tendency.
11. The typesetting of the quote should be the same as in the above. The font format of formulas (8), (9) and (10) is inconsistent with other formulas.
12. In Section 4.1, the description of the extrapolation procedures for modeling the linear frequency shift is too brief, please provide a detailed introduction.
Reviewer 2 Report
Authors have proposed Investigation of Material Loading on an Evolved Antecedent Hexagonal CSRR-Loaded Electrically Small Antenna. The following are observations/suggestions/recommendations:
1. Abstract: The abstract lacks the antenna parameters. Please put up the majority of the antenna parameters in the abstract. It may not be necessary to brief EM in the abstract.
2. The introduction should have all recent literature. The depth of the introduction is fine.
3. The figure 1 should be improved – it should be self-drawn images. The snapshots could be a little better. The same is applicable to Figure 2 and subsequent images.
4. The electrical dimensions of the antenna should be included in the manuscript to give an idea of the electrical compactness of the antenna.
5. Page 11, Line 513: “In addition to achieving a maximum return loss of 38.6 dB, there is an observable shift in resonant frequency after the designed antenna was subjected to material loading” – can you please scientifically justify the shift achieved through CSRR loading?
6. Figure 13 is poorly represented. Please refer to the above comments and improve all the images.
7. Did you try changing the dielectric constant to much higher values and see the effect?
8. All the equations must be specified in Math Editor.
9. The flow chart should be presented with a better representation.
Grammar should be verified across the manuscript.
Round 2
Reviewer 1 Report
The former questions are answered correspondingly. No further comments.
Minor editing of English language required.
Author Response
We thank the reviewer for the comment. Further amendments (see version with corrections highlighted) to the English language have been made to enhance the quality of the manuscript.
Reviewer 2 Report
The comments have been adequately addressed.
Author Response

(The authors gave the same response as above.)
